# Smooth and Strong:
# MAP Inference with Linear Convergence

**Ofer Meshi**
TTI Chicago

**Mehrdad Mahdavi**
TTI Chicago

**Alexander G. Schwing**
University of Toronto

## Abstract

Maximum a-posteriori (MAP) inference is an important task for many applications. Although the standard formulation gives rise to a hard combinatorial optimization problem, several effective approximations have been proposed and studied in recent years. We focus on linear programming (LP) relaxations, which have achieved state-of-the-art performance in many applications. However, optimization of the resulting program is in general challenging due to non-smoothness and complex non-separable constraints.

Therefore, in this work we study the benefits of augmenting the objective function of the relaxation with strong convexity. Specifically, we introduce strong convexity by adding a quadratic term to the LP relaxation objective. We provide theoretical guarantees for the resulting programs, bounding the difference between their optimal value and the original optimum. Further, we propose suitable optimization algorithms and analyze their convergence.

## 1 Introduction

Probabilistic graphical models are an elegant framework for reasoning about multiple variables with structured dependencies. They have been applied in a variety of domains, including computer vision, natural language processing, computational biology, and many more. Throughout, finding the *maximum a-posteriori* (MAP) configuration, *i.e.*, the most probable assignment, is one of the central tasks for these models. Unfortunately, in general the MAP inference problem is NP-hard. Despite this theoretical barrier, in recent years it has been shown that approximate inference methods based on *linear programming* (LP) relaxations often provide high quality MAP solutions in practice. Although tractable in principle, LP relaxations pose a real computational challenge. In particular, for many applications, standard LP solvers perform poorly due to the large number of variables and constraints [33]. Therefore, significant research effort has been put into designing efficient solvers that exploit the special structure of the MAP inference problem.

Some of the proposed algorithms optimize the primal LP directly, however this is hard due to complex coupling constraints between the variables. Therefore, most of the specialized MAP solvers optimize the dual function, which is often easier since it preserves the structure of the underlying model and facilitates elegant message-passing algorithms. Nevertheless, the resulting optimization problem is still challenging since the dual function is piecewise linear and therefore non-smooth. In fact, it was recently shown that LP relaxations for MAP inference are not easier than general LPs [22]. This result implies that there exists an inherent *trade-off* between the approximation error (accuracy) of the relaxation and its optimization error (efficiency).

In this paper we propose new ways to explore this trade-off. Specifically, we study the benefits of adding strong convexity in the form of a quadratic term to the MAP LP relaxation objective. We show that adding strong convexity to the primal LP results in a new smooth dual objective, which serves as an alternative to soft-max. This smooth objective can be computed efficiently and optimized via gradient-based methods, including accelerated gradient. On the other hand, introducing strong convexity in the dual leads to a new primal formulation in which the coupling constraints are enforced softly, through a penalty term in the objective. This allows us to derive an efficient

conditional gradient algorithm, also known as the Frank-Wolfe (FW) algorithm. We can then regularize both primal and dual to obtain a smooth and strongly convex objective, for which various algorithms enjoy linear convergence rate. We provide theoretical guarantees for the new objective functions, analyze the convergence rate of the proposed algorithms, and compare them to existing approaches. All of our algorithms are guaranteed to globally converge to the optimal value of the modified objective function. Finally, we show empirically that our methods are competitive with other state-of-the-art algorithms for MAP LP relaxation.

## 2  Related Work

Several authors proposed efficient approximations for MAP inference based on LP relaxations [e.g., 30]. Kumar et al. [12] show that LP relaxation dominates other convex relaxations for MAP inference. Due to the complex non-separable constraints, only few of the existing algorithms optimize the primal LP directly. Ravikumar et al. [23] present a proximal point method that requires iterative projections onto the constraints in the inner loop. Inexactness of these iterative projections complicates the convergence analysis of this scheme. In Section 4.1 we show that adding a quadratic term to the dual problem corresponds to a much easier primal in which agreement constraints are enforced softly through a penalty term that accounts for constraint violation. This enables us to derive a simpler projection-free algorithm based on conditional gradient for the primal relaxed program [4, 13]. Recently, Belanger et al. [1] used a different non-smooth penalty term for constraint violation, and showed that it corresponds to box-constraints on dual variables. In contrast, our penalty terms are smooth, which leads to a different objective function and faster convergence guarantees.

Most of the popular algorithms for MAP LP relaxations focus on the dual program and optimize it in various ways. The subgradient algorithm can be applied to the non-smooth objective [11], however its convergence rate is rather slow, both in theory and in practice. In particular, the algorithm requires $O(1/\epsilon^2)$ iterations to obtain an $\epsilon$-accurate solution to the dual problem. Algorithms based on coordinate minimization can also be applied [e.g., 6, 10, 31], and often converge fast, but they might get stuck in suboptimal fixed points due to the non-smoothness of the objective. To overcome this limitation it has been proposed to smooth the dual objective using a soft-max function [7, 8]. Coordinate minimization methods are then guaranteed to converge to the optimum of the smoothed objective. Meshi et al. [17] have shown that the convergence rate of such algorithms is $O(1/\gamma\epsilon)$, where $\gamma$ is the smoothing parameter. Accelerated gradient algorithms have also been successfully applied to the smooth dual, obtaining improved convergence rate of $O(1/\sqrt{\gamma\epsilon})$, which can be used to obtain a $O(1/\epsilon)$ rate w.r.t. the original objective [24]. In Section 4.2 we propose an alternative smoothing technique, based on adding a quadratic term to the primal objective. We then show how gradient-based algorithms can be applied efficiently to optimize the new objective function.

Other globally convergent methods that have been proposed include augmented Lagrangian [15, 16], bundle methods [9], and a steepest descent approach [25, 26]. However, the convergence rate of these methods in the context of MAP inference has not been analyzed yet, making them hard to compare to other algorithms.

## 3  Problem Formulation

In this section we formalize MAP inference in graphical models. Consider a set of $n$ discrete variables $X_1, \ldots, X_n$, and denote by $x_i$ a particular assignment to variable $X_i$. We refer to subsets of these variables by $r \subseteq \{1, \ldots, n\}$, also known as regions, and the total number of regions is referred to as $q$. Each subset is associated with a local score function, or factor, $\theta_r(x_r)$. The MAP problem is to find an assignment $x$ which maximizes a global score function that decomposes over the factors:

$$\max_x \sum_r \theta_r(x_r) \ .$$

The above combinatorial optimization problem is hard in general, and tractable only in several special cases. Most notably, for tree-structured graphs or super-modular pairwise score functions, efficient dynamic programming algorithms can be applied. Here we do not make such simplifying assumptions and instead focus on approximate inference. In particular, we are interested in approx-

imations based on the LP relaxation, taking the following form:

$$\max_{\mu \in \mathcal{M}_L} f(\mu) := \sum_r \sum_{x_r} \mu_r(x_r)\theta_r(x_r) \quad = \mu^\top \theta \tag{1}$$

$$\text{where:} \quad \mathcal{M}_L = \left\{ \mu \geq 0 \ \middle| \ \begin{array}{ll} \sum_{x_r} \mu_r(x_r) = 1 & \forall r \\ \sum_{x_p \backslash x_r} \mu_p(x_p) = \mu_r(x_r) & \forall r, x_r, p : r \in p \end{array} \right\},$$

where '$r \in p$' represents a containment relationship between the regions $p$ and $r$. The dual program of the above LP is formulated as minimizing the *re-parameterization* of factors [32]:

$$\min_\delta g(\delta) := \sum_r \max_{x_r} \left( \theta_r(x_r) + \sum_{p:r \in p} \delta_{pr}(x_r) - \sum_{c:c \in r} \delta_{rc}(x_c) \right) \equiv \sum_r \max_{x_r} \hat{\theta}_r^\delta(x_r), \tag{2}$$

This is a piecewise linear function in the dual variables $\delta$. Hence, it is convex (but not strongly) and non-smooth. Two commonly used optimization schemes for this objective are subgradient descent and block coordinate minimization. While the convergence rate of the former can be upper bounded by $O(1/\epsilon^2)$, the latter is non-convergent due to the non-smoothness of the objective function.

To remedy this shortcoming, it has been proposed to smooth the objective by replacing the local maximization with a soft-max [7, 8]. The resulting unconstrained program is:

$$\min_\delta g_\gamma(\delta) := \sum_r \gamma \log \sum_{x_r} \exp\left( \frac{\hat{\theta}_r^\delta(x_r)}{\gamma} \right). \tag{3}$$

This dual form corresponds to adding local entropy terms to the primal given in Eq. (1), obtaining:

$$\max_{\mu \in \mathcal{M}_L} \sum_r \sum_{x_r} \left( \mu_r(x_r)\theta_r(x_r) + \gamma H(\mu_r) \right), \tag{4}$$

where $H(\mu_r) = -\sum_{x_r} \mu_r(x_r) \log \mu_r(x_r)$ denotes the entropy. The following guarantee holds for the smooth optimal value $g_\gamma^*$:

$$g^* \leq g_\gamma^* \leq g^* + \gamma \sum_r \log V_r, \tag{5}$$

where $g^*$ is the optimal value of the dual program given in Eq. (2), and $V_r = |r|$ denotes the number of variables in region $r$.

The dual given in Eq. (3) is a smooth function with Lipschitz constant $L = \frac{1}{\gamma} \sum_r V_r$ [see 24]. In this case coordinate minimization algorithms are globally convergent (to the smooth optimum), and their convergence rate can be bounded by $O(1/\gamma\epsilon)$ [17]. Gradient-based algorithms can also be applied to the smooth dual and have similar convergence rate $O(1/\gamma\epsilon)$. This can be improved using Nesterov's acceleration scheme to obtain an $O(1/\sqrt{\gamma\epsilon})$ rate [24]. The gradient of Eq. (3) takes the simple form:

$$\nabla_{\delta_{pr}(x_r)} g_\gamma = \left( b_r(x_r) - \sum_{x_p \backslash x_r} b_p(x_p) \right), \quad \text{where} \quad b_r(x_r) \propto \exp\left( \frac{\hat{\theta}_r^\delta(x_r)}{\gamma} \right). \tag{6}$$

## 4 Introducing Strong Convexity

In this section we study the effect of adding strong convexity to the objective function. Specifically, we add the Euclidean norm of the variables to either the dual (Section 4.1) or primal (Section 4.2) function. We study the properties of the objectives, and propose appropriate optimization schemes.

### 4.1 Strong Convexity in the Dual

As mentioned above, the dual given in Eq. (2) is a piecewise linear function, hence not smooth. Introducing strong convexity to control the convergence rate, is an alternative to smoothing. We propose to introduce strong convexity by simply adding the $L_2$ norm of the variables to the dual

---

**Algorithm 1** Block-coordinate Frank-Wolfe for soft-constrained primal

---
1: Initialize: $\mu_r(x_r) = \mathbb{1}\{x_r = \text{argmax}_{x'_r}\, \hat{\theta}_r^{\delta(\mu)}(x'_r)\}$ for all $r, x_r$
2: **while** not converged **do**
3:     Pick $r$ at random
4:     Let $s_r(x_r) = \mathbb{1}\{x_r = \text{argmax}_{x'_r}\, \hat{\theta}_r^{\delta(\mu)}(x'_r)\}$ for all $x_r$
5:     Let $\eta = \dfrac{\left(\hat{\theta}_r^{\delta(\mu)}\right)^\top (s_r - \mu_r)}{\frac{1}{\lambda} P_r \|s_r - \mu_r\|^2 + \frac{1}{\lambda}\sum_{c:c\in r} \|A_{rc}(s_r - \mu_r)\|^2}$ , and clip to $[0,1]$
6:     Update $\mu_r \leftarrow (1-\eta)\mu_r + \eta s_r$
7: **end while**

---

program given in Eq. (2), *i.e.*,

$$\min_\delta \breve{g}_\lambda(\delta) := g(\delta) + \frac{\lambda}{2}\|\delta\|^2 \,. \tag{7}$$

The corresponding primal objective is then (see Appendix A):

$$\max_{\mu\in\Delta^\times} f_\lambda(\mu) := \mu^\top \theta - \frac{1}{2\lambda} \sum_{r,x_r,p:r\in p} \left(\sum_{x_p\backslash x_r} \mu_p(x_p) - \mu_r(x_r)\right)^2 \;=\; \mu^\top\theta - \frac{\lambda}{2}\|A\mu\|^2 \,, \tag{8}$$

where $\Delta^\times$ preserves only the separable per-region simplex constraints in $\mathcal{M}_L$, and for convenience we define $(A\mu)_{r,x_r,p} = \frac{1}{\lambda}\left(\sum_{x_p\backslash x_r} \mu_p(x_p) - \mu_r(x_r)\right)$. Importantly, this primal program is similar to the original primal given in Eq. (1), but the non-separable marginalization constraints in $\mathcal{M}_L$ are enforced softly – via a penalty term in the objective. Interestingly, the primal in Eq. (8) is somewhat similar to the objective function obtained by the steepest descent approach proposed by Schwing et al. [25], despite being motivated from different perspectives. Similar to Schwing et al. [25], our algorithm below is also based on conditional gradient, however ours is a single-loop algorithm, whereas theirs employs a double-loop procedure.

We obtain the following guarantee for the optimum of the strongly convex dual (see Appendix C):

$$g^* \le \breve{g}_\lambda^* \le g^* + \frac{\lambda}{2} h \,, \tag{9}$$

where $h$ is chosen such that $\|\delta^*\|^2 \le h$. It can be shown that $h = (4Mq\|\theta\|_\infty)^2$, where $M = \max_r W_r$, and $W_r$ is the number of configurations of region $r$ (see Appendix C). Notice that this bound is worse than the soft-max bound stated in Eq. (5) due to the dependence on the magnitude of the parameters $\theta$ and the number of configurations $W_r$.

**Optimization**    It is easy to modify the subgradient algorithm to optimize the strongly convex dual given in Eq. (7). It only requires adding the term $\lambda\delta$ to the subgradient. Since the objective is non-smooth and strongly convex, we obtain a convergence rate of $O(1/\lambda\epsilon)$ [19]. We note that coordinate descent algorithms for the dual objective are still non-convergent, since the program is still non-smooth. Instead, we propose to optimize the primal given in Eq. (8) via a conditional gradient algorithm [4]. Specifically, in Algorithm 1 we implement the block-coordinate Frank-Wolfe algorithm proposed by Lacoste-Julien et al. [13]. In Algorithm 1 we denote $P_r = |\{p : r \in p\}|$, we define $\delta(\mu)$ as $\delta_{pr}(x_r) = \frac{1}{\lambda}\left(\sum_{x_p\backslash x_r} \mu_p(x_p) - \mu_r(x_r)\right)$, and $A_{rc}\mu_r = \sum_{x_r\backslash x_c} \mu_r(x_r)$.

In Appendix D we show that the convergence rate of Algorithm 1 is $O(1/\lambda\epsilon)$, similar to subgradient in the dual. However, Algorithm 1 has several advantages over subgradient. First, the step-size requires no tuning since the optimal step $\eta$ is computed analytically. Second, it is easy to monitor the sub-optimality of the current solution by keeping track of the duality gap $\sum_r \left(\hat{\theta}_r^\delta\right)^\top (s_r - \mu_r)$, which provides a sound stopping condition.[1] Notice that the basic operation for the update is maximization over the re-parameterization $(\max_{x_r} \hat{\theta}_r^\delta(x_r))$, which is similar to a subgradient computation. This operation is sometimes cheaper than coordinate minimization, which requires computing max-

marginals [see 28]. We also point out that, similar to Lacoste-Julien et al. [13], it is possible to execute Algorithm 1 in terms of dual variables, without storing primal variables $\mu_r(x_r)$ for large parent regions (see Appendix E for details). As we demonstrate in Section 5, this can be important when using global factors.

We note that Algorithm 1 can be used with minor modifications in the inner loop of an augmented Lagrangian algorithm [15]. But we show later that this double-loop procedure is not necessary to obtain good results for some applications. Finally, Meshi et al. [18] show how to use the objective in Eq. (8) to obtain an efficient training algorithm for learning the score functions $\theta$ from data.

## 4.2 Strong Convexity in the Primal

We next consider appending the primal given in Eq. (1) with a similar $L_2$ norm, obtaining:

$$\max_{\mu \in \mathcal{M}_L} f_\gamma(\mu) := \mu^\top \theta - \frac{\gamma}{2} \|\mu\|^2 \,. \tag{10}$$

It turns out that the corresponding dual function takes the form (see Appendix B):

$$\min_\delta \tilde{g}_\gamma(\delta) := \sum_r \max_{u \in \Delta} \left( u^\top \hat{\theta}_r^\delta - \frac{\gamma}{2} \|u\|^2 \right) = \sum_r \left( \frac{\gamma}{2} \left\| \frac{\hat{\theta}_r^\delta}{\gamma} \right\|^2 - \min_{u \in \Delta} \frac{\gamma}{2} \left\| u - \frac{\hat{\theta}_r^\delta}{\gamma} \right\|^2 \right) \,. \tag{11}$$

Thus the dual objective involves scaling the factor reparameterization $\hat{\theta}_r^\delta$ by $1/\gamma$, and then projecting the resulting vector onto the probability simplex. We denote the result of this projection by $u_r$ (or just $u$ when clear from context). The $L_2$ norm in Eq. (10) has the same role as the entropy terms in Eq. (4), and serves to *smooth* the dual function. This is a consequence of the well known duality between strong convexity and smoothness [*e.g.*, 21]. In particular, the dual stated in Eq. (11) is smooth with Lipschitz constant $L = q/\gamma$.

To calculate the objective value we need to compute the projection $u_r$ onto the simplex for all factors. This can be done by sorting the elements of the scaled reparameterization $\hat{\theta}_r^\delta/\gamma$, and then shifting all elements by the same value such that all positive elements sum to 1. The negative elements are then set to 0 [see, *e.g.*, 3, for details]. Intuitively, we can think of $u_r$ as a max-marginal which does not place weight 1 on the maximum element, but instead spreads the weight among the top scoring elements, if their score is close enough to the maximum. The effect is similar to the soft-max case, where $b_r$ can also be thought-of as a soft max-marginal (see Eq. (6)). On the other hand, unlike $b_r$, our max-marginal $u_r$ will most likely be sparse, since only a few elements tend to have scores close to the maximum and hence non-zero value in $u_r$.

Another interesting property of the dual in Eq. (11) is invariance to shifting, which is also the case for the non-smooth dual provided in Eq. (2) and the soft-max dual given in Eq. (3). Specifically, shifting all elements of $\delta_{pr}(\cdot)$ by the same value does not change the objective value, since the projection onto the simplex is shift-invariant.

We next bound the difference between the smooth optimum and the original one. The bound follows easily from the bounded norm of $\mu_r$ in the probability simplex:

$$f^* - \frac{\gamma}{2}q \le f_\gamma^* \le f^* \,, \qquad \text{or equivalently:} \qquad f^* \le \left( f_\gamma^* + \frac{\gamma}{2}q \right) \le f^* + \frac{\gamma}{2}q \,.$$

We actually use the equivalent form on the right in order to get an upper bound rather than a lower bound.[2] From strong duality we immediately get a similar guarantee for the dual optimum:

$$g^* \le \left( \tilde{g}_\gamma^* + \frac{\gamma}{2}q \right) \le g^* + \frac{\gamma}{2}q \,.$$

Notice that this bound is better than the corresponding soft-max bound stated in Eq. (5), since it does not depend on the scope size of regions, *i.e.*, $V_r$.

|  | Convex | Strongly-convex |  |
|---|---|---|---|
| **Non-smooth** | **Dual:** $\min_\delta g(\delta) := \sum_r \max_{x_r} \hat{\theta}_r^\delta(x_r)$ <br> Subgradient $O(1/\epsilon^2)$ <br> CD (non-convergent) <br> ___ <br> **Primal:** $\max_{\mu \in \mathcal{M}_L} \mu^\top \theta$ <br> Proximal projections | **Dual:** $\min_\delta g(\delta) + \frac{\lambda}{2}\|\delta\|^2$ <br> Subgradient $O(1/\lambda\epsilon)$ <br> ___ <br> **Primal:** $\max_{\mu \in \Delta^\times} \mu^\top \theta - \frac{\lambda}{2}\|A\mu\|^2$ <br> FW $O(1/\lambda\epsilon)$ | Section 4.1 |
| **$L_2$-max** | **Dual:** $\min_\delta \tilde{g}_\gamma(\delta) := \sum_r \max_{u \in \Delta}\left(u^\top \hat{\theta}_r^\delta - \frac{\gamma}{2}\|u\|^2\right)$ <br> Gradient $O(1/\gamma\epsilon)$ <br> Accelerated $O(1/\sqrt{\gamma\epsilon})$ <br> CD? <br> ___ <br> **Primal:** $\max_{\mu \in \mathcal{M}_L} \mu^\top \theta - \frac{\gamma}{2}\|\mu\|^2$ <br> *(Section 4.2)* | **Dual:** $\min_\delta \tilde{g}_\gamma(\delta) + \frac{\lambda}{2}\|\delta\|^2$ <br> Gradient $O(\frac{1}{\gamma\lambda}\log(\frac{1}{\epsilon}))$ <br> Accelerated $O(\frac{1}{\sqrt{\gamma\lambda}}\log(\frac{1}{\epsilon}))$ <br> ___ <br> **Primal:** $\max_{\mu \in \Delta^\times} \mu^\top \theta - \frac{\lambda}{2}\|A\mu\|^2 - \frac{\gamma}{2}\|\mu\|^2$ <br> SDCA $O((1+\frac{1}{\gamma\lambda})\log(\frac{1}{\epsilon}))$ | Section 4.3 |
| **Soft-max** | **Dual:** $\min_\delta g_\gamma(\delta) := \sum_r \gamma \log \sum_{x_r} \exp\left(\frac{\theta_r^\delta(x_r)}{\gamma}\right)$ <br> Gradient $O(1/\gamma\epsilon)$ <br> Accelerated $O(1/\sqrt{\gamma\epsilon})$ <br> CD $O(1/\gamma\epsilon)$ <br> ___ <br> **Primal:** $\max_{\mu \in \mathcal{M}_L} \mu^\top \theta + \gamma \sum_r H(\mu_r)$ | **Dual:** $\min_\delta g_\gamma(\delta) + \frac{\lambda}{2}\|\delta\|^2$ <br> Gradient $O(\frac{1}{\gamma\lambda}\log(\frac{1}{\epsilon}))$ <br> Accelerated $O(\frac{1}{\sqrt{\gamma\lambda}}\log(\frac{1}{\epsilon}))$ <br> ___ <br> **Primal:** $\max_{\mu \in \Delta^\times} \mu^\top \theta - \frac{\lambda}{2}\|A\mu\|^2 + \gamma \sum_r H(\mu_r)$ | Section 4.3 |

Table 1: Summary of objective functions, algorithms and rates. Row and column headers pertain to the dual objective. Previously known approaches are shaded.

**Optimization**   To solve the dual program given in Eq. (11) we can use gradient-based algorithms. The gradient takes the form:

$$\nabla_{\delta_{pr}(x_r)}\tilde{g}_\gamma = \left(u_r(x_r) - \sum_{x_p \setminus x_r} u_p(x_p)\right),$$

which only requires computing the projection $u_r$, as in the objective function. Notice that this form is very similar to the soft-max gradient (Eq. (6)), with projections $u$ taking the role of beliefs $b$. The gradient descent algorithm applies the updates: $\delta \leftarrow \delta - \frac{1}{L}\nabla\tilde{g}_\gamma$ iteratively. The convergence rate of this scheme for our smooth dual is $O(1/\gamma\epsilon)$, which is similar to the soft-max rate [20]. As in the soft-max case, Nesterov's accelerated gradient method achieves a better $O(1/\sqrt{\gamma\epsilon})$ rate [see 24].

Unfortunately, it is not clear how to derive efficient coordinate minimization updates for the dual in Eq. (11), since the projection $u_r$ depends on the dual variables $\delta$ in a non-linear manner.

Finally, we point out that the program in Eq. (10) is very similar to the one solved in the inner loop of proximal point methods [23]. Therefore our gradient-based algorithm can be used with minor modifications as a subroutine within such proximal algorithms (requires mapping the final dual solution to a feasible primal solution [see, e.g., 17]).

### 4.3   Smooth and Strong

In order to obtain a smooth and strongly convex objective function, we can add an $L_2$ term to the smooth program given in Eq. (11) (similarly possible for the soft-max dual in Eq. (3)). Gradient-based algorithms have linear convergence rate in this case [20]. Equivalently, we can add an $L_2$ term to the primal in Eq. (8). Although conditional gradient is not guaranteed to converge linearly in this case [5], stochastic coordinate ascent (SDCA) does enjoy linear convergence, and can even be accelerated to gain better dependence on the smoothing and convexity parameters [27]. This requires only minor modifications to the algorithms presented above, which are highlighted in Appendix F. To conclude this section, we summarize all objective functions and algorithms in Table 1.

## 5   Experiments

We now proceed to evaluate the proposed methods on real and synthetic data and compare them to existing state-of-the-art approaches. We begin with a synthetic model adapted from Kolmogorov [10]. This example was designed to show that coordinate descent algorithms might get stuck in suboptimal points due to non-smoothness. We compare the following MAP inference algorithms: non-smooth coordinate descent (CD), non-smooth subgradient descent, smooth CD (for soft-max), gradient descent (GD) and accelerated GD (AGD) with either soft-max or $L_2$ smoothing (Section 4.2), our Frank-Wolfe Algorithm 1 (FW), and the linear convergence variants (Section 4.3). In Fig. 1

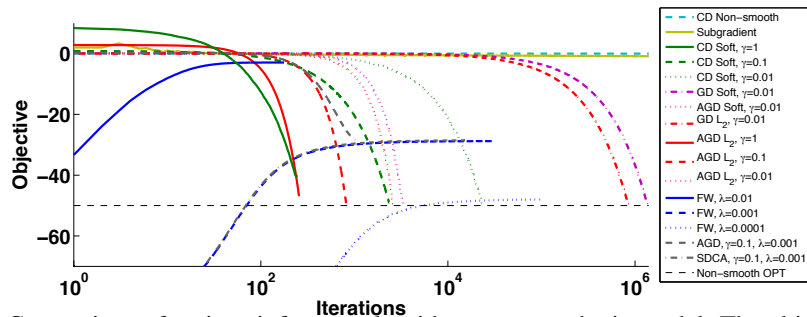

Figure 1: Comparison of various inference algorithms on a synthetic model. The objective value as a function of the iterations is plotted. The optimal value is shown in thin dashed dark line.

we notice that non-smooth CD (light blue, dashed) is indeed stuck at the initial point. Second, we observe that the subgradient algorithm (yellow) is extremely slow to converge. Third, we see that smooth CD algorithms (green) converge nicely to the smooth optimum. Gradient-based algorithms for the same smooth (soft-max) objective (purple) also converge to the same optimum, while AGD is much faster than GD. We can also see that gradient-based algorithms for the $L_2$-smooth objective (red) preform slightly better than their soft-max counterparts. In particular, they have faster convergence and tighter objective for the same value of the smoothing parameter, as our theoretical analysis suggests. For example, compare the convergence of AGD soft and AGD $L_2$ both with $\gamma = 0.01$. For the optimal value, compare CD soft and AGD $L_2$ both with $\gamma = 1$. Fourth, we note that the FW algorithm (blue) requires smaller values of the strong-convexity parameter $\lambda$ in order to achieve high accuracy, as our bound in Eq. (9) predicts. We point out that the dependence on the smoothing or strong convexity parameter is roughly linear, which is also aligned with our convergence bounds. Finally, we see that for this model the smooth and strongly convex algorithms (gray) perform similar or even slightly worse than either the smooth-only or strongly-convex-only counterparts.

In our experiments we compare the number of iterations rather than runtime of the algorithms since the computational cost per iteration is roughly the same for all algorithms (includes a pass over all factors), and the actual runtime greatly depends on the implementation. For example, gradient computation for $L_2$ smoothing requires sorting factors rather than just maximizing over their values, incurring worst-case cost of $O(W_r \log W_r)$ per factor instead of just $O(W_r)$ for soft-max gradient. However, one can use partitioning around a pivot value instead of sorting, yielding $O(W_r)$ cost in expectation [3], and caching the pivot can also speed-up the runtime considerably. Moreover, logarithm and exponent operations needed by the soft-max gradient are much slower than the basic operations used for computing the $L_2$ smooth gradient. As another example, we point out that AGD algorithms can be further improved by searching for the effective Lipschitz constant rather than using the conservative bound $L$ (see [24] for more details). In order to abstract away these details we compare the iteration cost of the vanilla versions of all algorithms.

We next conduct experiments on real data from a protein side-chain prediction problem from Yanover et al. [33]. This problem can be cast as MAP inference in a model with unary and pairwise factors. Fig. 2 (left) shows the convergence of various MAP algorithms for one of the proteins (similar behavior was observed for the other instances). The behavior is similar to the synthetic example above, except for the much better performance of non-smooth coordinate descent. In particular, we see that coordinate minimization algorithms perform very well in this setting, better than gradient-based and the FW algorithms (this finding is consistent with previous work [e.g., 17]). Only a closer look (Fig. 2, left, bottom) reveals that smoothing actually helps to obtain a slightly better solution here. In particular, the soft-max CD (with $\gamma = 0.001$) and $L_2$-max AGD (with $\gamma = 0.01$), as well as the primal (SDCA) and dual (AGD) algorithms for the smooth and strongly convex objective, are able to recover the optimal solution within the allowed number of iterations. The non-smooth FW algorithm also finds a near-optimal solution.

Finally, we apply our approach to an image segmentation problem with a global cardinality factor. Specifically, we use the Weizmann Horse dataset for foreground-background segmentation [2]. All images are resized to $150 \times 150$ pixels, and we use 50 images to learn the parameters of the model and the other 278 images to test inference. Our model consists of unary and pairwise factors along with a single global cardinality factor, that serves to encourage segmentations where the number of foreground pixels is not too far from the trainset mean. Specifically, we use the cardinality factor from Li and Zemel [14], defined as: $\theta_c(x) = \max\{0, |s - s_0| - t\}^2$, where $s = \sum_i x_i$. Here, $s_0$ is a reference cardinality computed from the training set, and $t$ is a tolerance parameter, set to $t = s_0/5$.

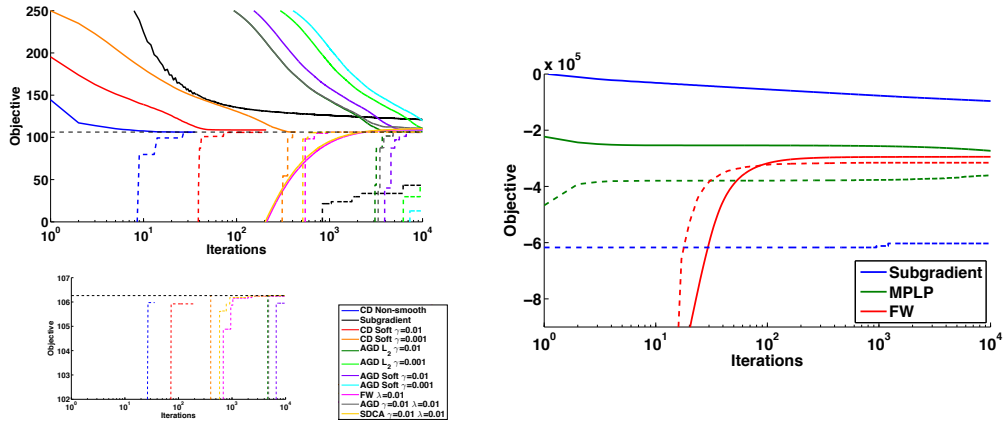

Figure 2: (Left) Comparison of MAP inference algorithms on a protein side-chain prediction problem. In the upper figure the solid lines show the optimized objective for each algorithm, and the dashed lines show the score of the best decoded solution (obtained via simple rounding). The bottom figure shows the value of the decoded solution in more detail. (Right) Comparison of MAP inference algorithms on an image segmentation problem. Again, solid lines show the value of the optimized objective while dashed lines show the score of the best decoded solution so far.

First we notice that not all of the algorithms are efficient in this setting. In particular, algorithms that optimize the smooth dual (either soft-max or $L_2$ smoothing) need to enumerate factor configurations in order to compute updates, which is prohibitive for the global cardinality factor. We therefore take the non-smooth subgradient and coordinate descent [MPLP, 6] as baselines, and compare their performance to that of our FW Algorithm 1 (with $\lambda = 0.01$). We use the variant that does not store primal variables for the global factor (Appendix E). We point out that MPLP requires calculating max-marginals for factors, rather than a simple maximization for subgradient and FW. In the case of cardinality factors this can be done at similar cost using dynamic programming [29], however there are other types of factors where max-marginal computation might be more expensive than max [28].

In Fig. 2 (right) we show a typical run for a single image, where we limit the number of iterations to $10K$. We observe that subgradient descent is again very slow to converge, and coordinate descent is also rather slow here (in fact, it is not even guaranteed to reach the optimum). In contrast, our FW algorithm converges orders of magnitude faster and finds a high quality solution (for runtime comparison see Appendix G). Over the entire 278 test instances we found that FW gets the highest score solution for 237 images, while MPLP finds the best solution in only 41 images, and subgradient never wins. To explain this success, recall that our algorithm enforces the agreement constraints between factor marginals only softly. It makes sense that in this setting it is not crucial to reach full agreement between the cardinality factor and the other factors in order to obtain a good solution.

## 6  Conclusion

In this paper we studied the benefits of strong convexity for MAP inference. We introduced a simple $L_2$ term to make either the dual or primal LP relaxations strongly convex. We analyzed the resulting objective functions and provided theoretical guarantees for their optimal values. We then proposed several optimization algorithms and derived upper bounds on their convergence rates. Using the same machinery, we obtained smooth and strongly convex objective functions, for which our algorithms retained linear convergence guarantees. Our approach offers new ways to trade-off the approximation error of the relaxation and the optimization error. Indeed, we showed empirically that our methods significantly outperform strong baselines on problems involving cardinality potentials.

To extend our work we aim at natural language processing applications since they share characteristics similar to the investigated image segmentation task. Finally, we were unable to derive closed-form coordinate minimization updates for our $L_2$-smooth dual in Eq. (11). We hope to find alternative smoothing techniques which facilitate even more efficient updates.

## Footnotes

[1]Similar rate guarantees can be derived for the duality gap.

[2]In our experiments we show the shifted objective value.

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
