[Supplementary Material]

# Supplementary Material
## Smooth and Strong: MAP Inference with Linear Convergence

## A   The Primal of the Strongly-Convex Dual

We start from the strongly convex dual Eq. (7):

$$\min_{\delta} \frac{\lambda}{2}\|\delta\|^2 + \sum_r \max_{x_r} \left( \theta_r(x_r) + \sum_{p:r\in p} \delta_{pr}(x_r) - \sum_{c:c\in r} \delta_{rc}(x_c) \right)$$

To derive the primal we rewrite this dual as:

$$\min_{\delta,\xi} \frac{\lambda}{2}\|\delta\|^2 + \sum_r \xi_r$$

$$\text{s.t. } \xi_r \geq \theta_r(x_r) + \sum_{p:r\in p} \delta_{pr}(x_r) - \sum_{c:c\in r} \delta_{rc}(x_c) \quad \forall r, x_r$$

The Lagrangian is then:

$$L(\delta, \xi, \mu \geq 0) = \frac{\lambda}{2}\|\delta\|^2 + \sum_r \xi_r$$

$$+ \sum_r \sum_{x_r} \mu_r(x_r) \left[ \theta_r(x_r) + \sum_{p:r\in p} \delta_{pr}(x_r) - \sum_{c:c\in r} \delta_{rc}(x_c) - \xi_r \right]$$

Deriving optimality conditions, we obtain:

$$\frac{\partial L}{\partial \xi_r} = 1 - \sum_{x_r} \mu_r(x_r) = 0 \qquad \text{[normalization]}$$

$$\frac{\partial L}{\partial \delta_{pr}(x_r)} = \lambda \delta_{pr}(x_r) - \mu_p(x_r) + \mu_r(x_r) = 0 \qquad \Rightarrow \delta_{pr}(x_r) = \frac{1}{\lambda} \left( \mu_p(x_r) - \mu_r(x_r) \right)$$

where $\mu_p(x_r) = \sum_{x_p \setminus x_r} \mu_p(x_p)$.
Plugging back in $L$, we get the primal:

$$\max_{\mu \in \Delta^\times} \mu^\top \theta - \frac{\lambda}{2}\|A\mu\|^2$$

## B   $L_2$-Smoothed Dual

We first show the $L_2$ smoothing for the max function and then use it to derive the dual. The following derivation is based on section (5.7.1) in Boyd and Vandenberghe (2004) and section (5.1) in Shalev-Shwartz and Zhang (2014).

$$\phi(x) = \max_i x_i = \max_{\beta \in \Delta} \sum_i \beta_i x_i$$

$$\phi^\star(y) = \max_x x^\top y - \phi(x) = \max_x x^\top y - \max_{\beta \in \Delta} \beta^\top x = \min_{\beta \in \Delta} \max_x (y - \beta)^\top x$$

where $\phi^\star$ is the convex conjugate of $\phi$. Unless $y = \beta$ the inner max gives $\infty$. Therefore:

$$\phi^\star(y) = \begin{cases} 0 & y \in \Delta \\ \infty & \text{ow} \end{cases}$$

To smooth, we add a simple $L_2$ term:

$$\phi_\gamma^\star(y) = \begin{cases} \frac{\gamma}{2}\|y\|^2 & y \in \Delta \\ \infty & \text{ow} \end{cases}$$

Going back to the primal:

$$\phi_\gamma(x) = \max_y x^\top y - \phi_\gamma^\star(y) = \max_{y \in \Delta} x^\top y - \frac{\gamma}{2}\|y\|^2$$

From this we obtain the bound:

$$\max_i x_i \leq \frac{\gamma}{2} + \max_{u \in \Delta} \left( u^\top x - \frac{\gamma}{2}\|u\|^2 \right)$$

We start from the smoothed dual and show the corresponding primal has the desired form.

$$\min_\delta \sum_r \max_{u \in \Delta} \left( u^\top (\theta_r + A_r^\top \delta) - \frac{\gamma}{2}\|u\|^2 \right) \tag{12}$$

where $A_r^\top$ is the reparameterization matrix (and $A_r$ is a marginalization matrix, but without the scaling $1/\lambda$). Next, we rewrite this by introducing auxiliary variables and equality constraints:

$$\min_{z,\delta} \sum_r \max_{u \in \Delta} \left( u^\top z_r - \frac{\gamma}{2}\|u\|^2 \right)$$

$$\text{s.t. } z_r(x_r) = \theta_r(x_r) + A_{r,x_r}^\top \delta \text{ for all } r, x_r$$

The Lagrangian is then:

$$L(z, \delta, \mu) = \tilde{\phi}_\gamma(z) + \mu^\top (\theta + A^\top \delta - z)$$

where $\tilde{\phi}_\gamma(z) = \sum_r \phi_\gamma(z_r)$. Hence the dual is:

$$\max_\mu \min_{z,\delta} \tilde{\phi}_\gamma(z) + \mu^\top (\theta + A^\top \delta - z)$$

$$= \max_\mu \mu^\top \theta + \underbrace{(\min_z \tilde{\phi}_\gamma(z) - \mu^\top z)}_{-\tilde{\phi}_\gamma^\star(\mu)} + (\min_\delta \delta^\top A\mu)$$

Unless $A\mu = 0$, the $\min_\delta$ term goes to $-\infty$, so we get:

$$\max_\mu \mu^\top \theta - \tilde{\phi}_\gamma^\star(\mu) \qquad \text{s.t. } A\mu = 0$$

Notice that $A\mu = 0$ enforces marginalization constraints.

Now, since $\tilde{\phi}_\gamma(z) = \sum_r \phi_\gamma(z_r)$ (each term in the sum has its own exclusive variables $z_r$), the dual is the sum of duals $\tilde{\phi}_\gamma^\star(\mu) = \sum_r \phi_\gamma^\star(\mu_r)$. So finally plugging $\phi_\gamma^\star(\mu)$ from before we get the primal:

$$\max_{\mu \in \mathcal{M}_L} \mu^\top \theta - \frac{\gamma}{2}\|\mu\|^2$$

as required.

## C  Guarantees for Strongly-Convex Dual

Denote by $g^*$ the optimal value of $g(\delta)$, and $\delta^*$ is a minimizer. We similarly define $g_\lambda^*$ and $\delta_\lambda^*$. We have:

$$
\begin{aligned}
g^* &= \min_\delta g(\delta) \\
&\leq g(\delta_\lambda^*) \\
&\leq g(\delta_\lambda^*) + \frac{\lambda}{2}\|\delta_\lambda^*\|^2 \\
&= g_\lambda^*
\end{aligned}
$$

On the other hand, we use a bound on the norm of the optimal solution $\|\delta^*\|^2 \leq h$, where $h = (4Mq\|\theta\|_\infty)^2$ (see Appendix B.1 in Meshi et al. (2015)).

$$
\begin{aligned}
g_\lambda^* &= \min_\delta g(\delta) + \frac{\lambda}{2}\|\delta\|^2 \\
&\leq g(\delta^*) + \frac{\lambda}{2}\|\delta^*\|^2 \\
&= g^* + \frac{\lambda}{2}\|\delta^*\|^2 \\
&\leq g^* + \frac{\lambda}{2}h
\end{aligned}
$$

So finally we obtain:

$$
g^* \leq g_\lambda^* \leq g^* + \frac{\lambda}{2}h
$$

Note that this proof still holds if we use the smooth $g_\gamma$ instead of $g$.

## D  Convergence Rate of Algorithm 1

To analyze the convergence rate of Algorithm 1, we use the following result from Lacoste-Julien et al. [13]:

**Theorem D.1.** *For each $t > 0$ it holds that $f_\lambda^* - \mathbb{E}\left[f_\lambda^{(t)}\right] \leq \frac{2q}{t+2q}\left(C_{f_\lambda}^\otimes + (f_\lambda^* - f_\lambda^{(0)})\right)$, where $C_{f_\lambda}^\otimes$ is the curvature constant of $f_\lambda$, and the expectation is over the random choice of factors. Furthermore, the duality gap is bounded by: $\mathbb{E}\left[D_\lambda^{(\hat{t})}\right] \leq \frac{6q}{t+1}\left(C_{f_\lambda}^\otimes + (f_\lambda^* - f_\lambda^{(0)})\right)$, for some $0 \leq \hat{t} \leq t$.*

To use this result, we need to bound the curvature constant of the objective function, denoted by $C_{f_\lambda}^\otimes$. It is shown in Lacoste-Julien et al. [13] that in the case of a product domain, the overall curvature is the sum of block-wise curvature constants: $C_{f_\lambda}^\otimes = \sum_r C_{f_\lambda}^r$

Furthermore, the curvature constant of a single factor is bounded in terms of the Hessian as follows:

$$
C_{f_\lambda}^r \leq \sup_{\substack{\mu,\mu' \in \Delta^\times, \\ (\mu'-\mu) \in \Delta_r, \\ z \in [\mu,\mu'] \subseteq \Delta^\times}} (\mu' - \mu)^\top \nabla^2 f(z)(\mu' - \mu)\,,
$$

To use this bound, we compute the Hessian for our problem[3] Eq. (8): $\nabla^2_\mu = \lambda A^\top A$, which is constant in $\mu$. Using arguments similar to Lemma A.2 in Lacoste-Julien et al. [13], we obtain:

$$
\begin{aligned}
C^r_{f_\lambda} &\leq \sup_{\substack{\mu,\mu' \in \Delta^\times, \\ (\mu'-\mu) \in \Delta_r}} (\mu' - \mu)^\top \left(\lambda A^\top A\right)(\mu' - \mu) \\
&\leq \lambda \sup_{\substack{\mu,\mu' \in \Delta^\times, \\ (\mu'-\mu) \in \Delta_r}} \|A(\mu' - \mu)\|_2^2 \\
&\leq 4\lambda \sup_{v \in A\Delta_r} \|v\|_2^2 \\
&\leq \frac{4R^2}{\lambda}
\end{aligned}
$$

where $R = 1 + \max_{p,r} \frac{W_p}{W_r}$ is the maximal number of marginalized assignments.

Finally, we have:

$$
C^\otimes_{f_\lambda} = \sum_r C^r_{f_\lambda} \leq q \left(\frac{4R^2}{\lambda}\right) = O\left(\frac{q}{\lambda}\right)
$$

Plugging this in Theorem D.1, we obtain a rate of $O(\frac{q^2}{\lambda\epsilon})$ for both the objective and duality gap.

## E   Executing Algorithm 1 with Global Factors

To execute Algorithm 1 for global factors, we need to avoid storing $\mu_r$ for the large factors. We assume those factors have no parents in the region graph, which is sensible. We will maintain the following variables during the run:

$$
\begin{aligned}
\delta_{pr}(x_r) &= \frac{1}{\lambda}\left(\mu_p(x_r) - \mu_r(x_r)\right) && \forall r, x_r, p : r \in p \\
\alpha_r &= \mu_r^\top \theta_r && \forall r \\
\beta_{rc}(x_c) &= \sum_{x_r \backslash x_c} \mu_r(x_r) && \forall r, c : c \in r
\end{aligned}
$$

Notice that we never store $\beta$ variables for regions with no parents (e.g., global).

Now, we explain how Algorithm 1 can be executed when the chosen region $r$ has global scope. First, $\hat\theta^\delta_r$ can be easily computed from the maintained dual variables $\delta$, and then $s_r$ can be obtained by maximization, which is assumed tractable. Second, to compute the step size $\eta$ we need for the nominator:[4]

$$
\begin{aligned}
\hat\theta_r^\top (s_r - \mu_r) &= \hat\theta_r^\top s_r - \hat\theta_r^\top \mu_r \\
&= \hat\theta_r^\top s_r - \theta_r^\top \mu_r - \mu_r^\top (A_r^\top \delta_r) \\
&= \hat\theta_r^\top s_r - \theta_r^\top \mu_r + \sum_c \delta_{rc}^\top \beta_{rc}
\end{aligned}
$$

where $\delta_r = \delta_{r\cdot}(\cdot)$, and $A_r$ is the reparameterization matrix, as before. For the denominator, the first term is 0 since $P_r = 0$. Furthermore, $\sum_{x_{r\backslash c}} s_r(x_r)$ is easy to compute since $s_r$ is an indicator for the maximizer, and $\beta_{rc}(x_c) = \sum_{x_{r\backslash c}} \mu_r(x_r)$ is maintained. Finally, the updates to the auxiliary

variables are:

$$\delta_{rc}(x_c) \leftarrow \delta_{rc}(x_c) + \frac{\eta}{\lambda} \sum_{x_r \setminus x_c} s_r(x_r) - \frac{\eta}{\lambda} \beta_{rc}(x_c) \qquad \forall c : c \in r$$

$$\alpha_r \leftarrow (1-\eta)\alpha_r + \eta \theta_r^\top s_r$$

$$\beta_{rc}(x_c) \leftarrow (1-\eta)\beta_{rc}(x_c) + \eta \sum_{x_r \setminus x_c} s_r \qquad \forall c : c \in r$$

## F  Algorithms with Linear Convergence Rate

In this section we describe the objective functions and algorithms for the smooth and strongly convex variants (the difference from the basic formulation is marked in blue).

First, we add an $L_2$ term to the smooth dual Eq. (11), obtaining:

$$\min_\delta \bar{g}_{\gamma,\lambda} := \tilde{g}_\gamma(\delta) + \frac{\lambda}{2}\|\delta\|^2 \tag{13}$$

Combining the guarantees of sections 4.1 and 4.2 yields:

$$g^* \leq \bar{g}^*_{\gamma,\lambda} \leq g^* + \frac{\lambda}{2}h + \frac{\gamma}{2}q$$

The gradient then needs to be slightly adjusted:

$$\nabla_{\delta_{pr}(x_r)} \bar{g}_{\gamma,\lambda} = \left( u_r(x_r) - \sum_{x_p \setminus x_r} u_p(x_p) \right) + \lambda \delta_{pr}(x_r)$$

Hence we can run gradient-based algorithms with this small modification.

As expected, the primal function corresponding to the smooth and strongly convex dual of Eq. (13) is equivalent to appending an $L_2$ term to the primal in Eq. (8):

$$\max_{\mu \in \Delta^\times} \mu^\top \theta - \frac{\lambda}{2}\|A\mu\|^2 - \frac{\gamma}{2}\|\mu\|^2$$

To apply dual coordinate ascent [27] (with line search), we make the following modifications in Algorithm 1. First, we replace $s_r$ by the projection onto the simplex $u_r$. Second, we use $\tilde{\theta}_r^\delta(x_r) = \hat{\theta}_r^\delta(x_r) - \gamma \mu_r(x_r)$ instead of $\hat{\theta}_r^\delta(x_r)$ when computing the step size:

$$\eta = \frac{\tilde{\theta}_r^\top (u_r - \mu_r)}{(\gamma + \frac{1}{\lambda}P_r)\|u_r - \mu_r\|^2 + \frac{1}{\lambda}\sum_{c:c \in r}\|A_{rc}(u_r - \mu_r)\|^2} \ .$$

Finally, we point out that this algorithm cannot be used with global factors due to the need to store $u_r$.

## G  Runtime Comparison

In this section we compare the runtime of various MAP inference algorithms for the image segmentation problem from Section 5. We show convergence on time scale rather than iteration scale in Fig. 3, which is analogous to Fig. 2 (right). We can see that the behavior is very similar to Fig. 2 since the cost of an iteration is roughly the same for all algorithms. For MPLP we use the efficient message computation from Tarlow et al. (2010). In fact, FW shows slightly better runtime, but the main reason is the faster computation of the objective value. Computing the objective is faster because we use the FW variant from Appendix E, which maintains dual variables that make this computation very fast.

Figure 3: Same as Fig. 2 (right), but the objective values are shown as a function of runtime instead of iterations.

## Footnotes

[3]Here we actually use the *negative* of Eq. (8) and treat this as a minimization problem.

[4]To simplify notation we use $\hat\theta_r$ instead of $\hat\theta^\delta_r$.