[Reviews · NeurIPS 2015]

Submitted by Assigned_Reviewer_1

Overview: This paper studies the benefits of augmenting the linear programming relaxation of the maximum a-posteriori (MAP) inference problem in graphical models with a quadratic term, thereby achieving strong convexity.

Such augmented formulations are obtained both from the original primal and dual formulations, and in each case the resulting primal-dual relationship is studied.

Prior work has mostly focused on smoothing the LP formulation using a softmax/entropy term, with a few notable exceptions, such as [5], [17] and [18].

Rather than those previous approaches, which employ a quadratic term in the sub-problems of either a *proximal* or a *alternating direction* scheme, in the present manuscript, the quadratic smoothing term is added directly.

This can in some way be seen as a naive approach: In comparison to proximal or alternating direction schemes, convergence to the global optimum of the original problem is no longer guaranteed, and the approximation quality directly depends on the strength of the augmentation term.

The aforementioned approaches adjust the strength in an adaptive manner and thus find the global optimum, at least in the limit.

However, the formulations derived from this direct augmentation are very insightful and reveal several interesting connections to other formulations.

Moreover, the authors demonstrate in several experiments (both synthetic and real-world), that a particular optimization scheme for the primal form of the augmented dual (based on block-coordinate Frank-Wolfe) compares very favorably to other practical schemes if a commensurate strength of the quadratic penalty is chosen.

The Frank-Wolfe method has recently seen numerous useful applications in machine learning, and this is yet another interesting example.

Positive points: + The paper is exceptionally well-written and organized. Related work is cited and the authors do an excellent job at connecting the various existing approaches and putting them into perspective. + The suggested optimization scheme (FW) seems practical.

It is reasonably efficient and has the advantage that only maximization (as opposed to max-marginalization, or even softmax-marginalization) is needed, which allows for the use of certain structured potentials. + The experimental evaluation is unusually thorough for a paper of this type, involving even inference experiments based on real-world learned structured prediction models. + The paper seems to be technically correct as far as I can tell. + Some of the derivations are novel to my knowledge, and the established optimization scheme employes the recently suggested block-coordinate Frank-Wolfe scheme for simplex-constrained quadratic optimization in a novel context. + Table 1 is extremely helpful in connecting the dots.

Negative points: - From a convex optimization point of view, the proposed scheme (FW) is less advanced than either of [5], [17], and [18]. - In fact, as the authors point out, their augmented formulation is also not entirely novel, but very closely related to the sub-problem in [17] (or the one in [5], if the primal is directly augmented).

As such, some of the derivations and bounds in the manuscript also build on material developed in previous work.

A comment: The quadratic programming relaxation suggested by Kumar [4] is also a simplex-constrained convex quadratic program.

It might be interesting to extend the discussion in the paper to highlight which, if any, similarities exist between the soft-constrained primal formulation suggested in this manuscript and Kumar's convex QP relaxation - in particular from an approximation quality point of view. (Note that by adding a quadratic augmentation term, as suggested in the present manuscript, the approximation also becomes one that differs from the original LP relaxation.)
Summary: This is a very well-written paper that makes strong contributions to the literature on approximate MAP inference in graphical models.

The contributions are two-fold: First, the manuscript helps consolidate and connect the vast number of existing approaches; and second, a novel approach to MAP inference/optimization scheme is suggested that seems to have a lot of merit in itself.

Submitted by Assigned_Reviewer_2

This paper introduces three different algorithms for discrete MAP inference based on the LP relaxation. By adding small terms to the primal, the dual or both, smoothness, strong convexity, or both can be obtained. Smoothness avoids getting stuck in local optima. Convergence properties and bounds on the original objective function are derived.

The paper is very well-written and well organized. It introduces and characterizes each of the variations of the LP, describing their advantages and shortcomings in great detail and with proper references to the literature.

I like the summary in Table 1. However, when consulted directly, it is unclear whether the smoothness/strongly-convex properties apply to the primal or the dual. For instance, when looking at the cell corresponding to 4.2, the table assigns it to the category "convex", but the primal is actually "strongly convex".

The experimental section is reasonably complete and shows the advantages and disadvantages of a number of methods and parameterizations, illustrating where the proposed approaches excel or fail in relation with existing literature. The need for MAP inference is pervasive, so this paper is potentially very significant (depending on how well the proposed algorithms perform on a wider array of problems).

Something that was not commented in the paper is in which range the introduced parameters (lambda, gamma) are actually usable, since I assume that setting very small values for these parameters results in accumulated rounding errors.
Summary: This work provides variations of the LP relaxation for MAP inference that favor the convergence of different types of optimizers. Good paper with exhaustive references, clear and complete exposition and adequate experimental validation.

Submitted by Assigned_Reviewer_3

Most relaxation-based methods for inference in graphical models work by relaxing the problem of inference in graphical models to convex constrained non-smooth optimization problems, which are then solved, rounded, and whose results are used in downstream applications. Since solving convex non-smooth optimization problems is hard (most existing algorithms either have poor convergence rates or poor computational complexity), this paper proposes smoothing down the problems by adding quadratic terms to either the primal or dual formulation. These lead to (some more, some less) natural applications of existing optimization algorithms which are known to have fast convergence rates at acceptable computational complexity. The paper evaluates these new methods on some models, obtaining promising results in terms of the value of the optimization function.

The main issue with this paper is that while the results seem correct (and indeed are quite straightforward to rederive) the main point of doing inference in graphical models is to use the output of inference as a part of some downstream task. Most relaxation-based methods make this quite easy as rounding is relatively straightforward except in some special cases where the relaxation is not tight (and even then approximate rounding is used).

Adding a quadratic term to either the primal or the dual inference objective will almost always lead to fractional solutions instead of integral solutions, making rounding harder, and almost arbitrary. Of course it's possible to do something similar to interior point methods and slowly anneal down the smoothness terms until you effectively have an integral solution, but this is not what this paper does.

Finally, there is already a relatively popular relaxation-based inference method which adds smoothness, reference [17] in the paper, Martins et al's AD3. It'd be very interesting to see comparisons between AD3 and this paper's method, as it's a closer comparison.

line 313 "Stochastic coordinate ascent" should read "stochastic dual coordinate ascent"

---- Revised after author response

After thinking more about this, reading the author responses, and discussing with the other reviewers I'm reviewing my score upwards, as I think that using these algorithms as subroutines in other solvers can be useful, and the primal solutions do see to be quite good.
Summary: The paper has an interesting idea for how to speed up convergence of LP relaxations for inference in graphical models at the expense of having a less useful result.

Submitted by Assigned_Reviewer_4

By adding regularisation to either the dual or the primal LP relaxation problem, strong convexity is achieved which in turn leads to optimisation algorithm that converge faster. Extensive comparisons of various optimisation (/inference) schemes

is provided.
Summary: MAP inference is an important problem and this work is a significant contribution towards faster LP inference schemes. The sectioning is reasonable, the paper well written (albeit it could be improved, i.e. experimental section).

Submitted by Assigned_Reviewer_5

The authors describe a set of optimization methods for the LP relaxation of MAP, based on various quadratic (and thus strongly convex) smoothing mechanisms in either the dual or primal formulations of the problem.

Strong convexity allows them to give fast asymptotic convergence rates for gradient and accelerated gradient optimization, at the cost of optimizing a slightly different objective function, whose difference they bound from the original.

The paper is very well written, with relatively good coverage of the many existing approaches to this problem and a clear description of how the authors' method differs from them.

The authors' approach seems novel to the best of my knowledge.

The experiments show the approach compares favorably with standard coordinate descent and subgradient descent methods, as well as producing better estimates with simple local decoding.

The paper does have a few weaknesses.

The LP bounds on MAP are very well studied, and most of the concepts used here have been applied previously in slightly different ways.

For example (as they discuss briefly) the authors' approach is very closely related to proximal methods and augmented Lagrangian methods, although they study single-loop, smoothed objectives rather than double-loop forms.

One positive is the careful analysis of various convex & strongly convex smoothing options (e.g., Table 1), which I think gives a useful taxonomy of methods.

However I am not entirely convinced that the methods, or their relative convergence rates, have a significant practical impact.

At a subjective level, they behave much the same way other smooth, globally convergent methods do.

With regards to the improvements in local decoding, this is a fairly simple baseline compared to decoding using search or branch-and-cut methods, or even a simple greedy decoding or a small amount of local search.

Overall I think the paper is very well written and would like to see it published.

I gave it a slightly lower score because I did not find the results particularly exciting and despite good experimental assessment of the paper's claims, I am not sure if they will turn out to be practically useful.
Summary: A very well written paper on several quadratically smoothed variations on the LP relaxation for MAP.

The quality and clarity are excellent, but it is very closely related to many existing approaches, and might not necessarily have a significant impact in practice.

Author Feedback
Author rebuttal: We thank the reviewers for the positive feedback.

Reviewers 1, 3 and 6:
- Connection to proximal methods and ADMM: As mentioned in Section 4, a more accurate solution can be obtained by using our method in the inner loop of a proximal method, ADMM, or an annealing procedure. We believe that this fact actually makes our contribution stronger. First, we show how to solve the problems in the inner loop of these algorithms more efficiently and provide new insights into the resulting problems via primal-dual relationships. Second, we show that in some applications a more relaxed objective function is sufficient for obtaining a high-quality solution, but with a significant computational gain by avoiding the outer loop.

Reviewers 4 and 6:
- Regarding lack of guarantees for a decoded/rounded solution, we would like to point out that due to the inherent hardness of the underlying combinatorial problem, the quality of a rounded solution can only be guaranteed under additional conditions on the problem, for example pairwise factors and certain potentials (see further discussion in [4] and [5]). Although it is interesting to study our approach in these special cases, we choose to keep the presentation as general as possible. We feel that such exploration is beyond the scope of the current paper, and perhaps more appropriate for an extended journal version. We do provide some empirical evidence on the quality of rounded solutions in the paper. In particular, in our experiments (Fig. 2, left and right) we show the value of a decoded primal solution (dashed lines) along with the optimized objective. These results demonstrate that our rounded solutions are very competitive with those obtained by the baseline algorithms.

Reviewer 1:
- Connection of our approach to the formulation in Kumar et al. [4]: It seems to us that although one of our proposed objective functions (Eq. (8)) is also a QP with local simplex constraints, the formulations are actually quite different. First, the variables in [4] are indicators over singleton assignments, and pairwise marginals are not even represented explicitly, which is somewhat different from our pseudo-marginals (mu). Second, the potentials used in the objective function of [4] are modified such that the quadratic term corresponds to pairwise scores, and they also ensure that the matrix is PSD (so the QP is convex). In contrast, in our objective the linear term contains the original potentials (singleton, pairwise, and high-order), and the quadratic term corresponds to violation of marginalization constraints (which are implicit in [4]). Therefore, it currently seems hard to draw direct connections between these two forms and compare their approximation quality.

Reviewer 2:
- In Tab. 1, the terminology pertains to the dual objective. We apologize for the confusion and will clarify in a revised version.
- Regarding the range of useful smoothness/convexity parameters: In our experiments we found that our approach was not very sensitive to the precise values, and any choice in the rage 0.1-0.0001 gave a reasonable trade-off between accuracy and runtime. We will add a comment in the revision.

Reviewer 3:
- Regarding practical impact we emphasize that among algorithms based on factor maximization (as opposed to more involved oracles discussed, e.g., in [20]), our FW algorithm (Alg 1) is a significant addition to the limited number of convergent alternatives to subgradient. As some reviewers point out and as we also demonstrate in the experiments, this may be very important in applications employing certain (high-order) structured potentials, especially since subgradient may be very slow in practice. In particular, we have shown its usefulness in a common vision task (segmentation). Similar potentials are also very common in NLP applications (e.g., parsing).